# Benchmarking RNA Editing Detection Tools

**DOI:** 10.3390/biotech12030056

**Published:** 2023-08-26

**Authors:** David Rodríguez Morales, Sarah Rennie, Shizuka Uchida

**Affiliations:** 1Department of Biology, University of Copenhagen, DK-2200 Copenhagen N, Denmark; davidtesorillo@hotmail.com; 2Center for RNA Medicine, Department of Clinical Medicine, Aalborg University, DK-2450 Copenhagen SV, Denmark

**Keywords:** databases, epitranscriptomics, RNA editing, RNA sequencing, tools

## Abstract

RNA, like DNA and proteins, can undergo modifications. To date, over 170 RNA modifications have been identified, leading to the emergence of a new research area known as epitranscriptomics. RNA editing is the most frequent RNA modification in mammalian transcriptomes, and two types have been identified: (1) the most frequent, adenosine to inosine (A-to-I); and (2) the less frequent, cysteine to uracil (C-to-U) RNA editing. Unlike other epitranscriptomic marks, RNA editing can be readily detected from RNA sequencing (RNA-seq) data without any chemical conversions of RNA before sequencing library preparation. Furthermore, analyzing RNA editing patterns from transcriptomic data provides an additional layer of information about the epitranscriptome. As the significance of epitranscriptomics, particularly RNA editing, gains recognition in various fields of biology and medicine, there is a growing interest in detecting RNA editing sites (RES) by analyzing RNA-seq data. To cope with this increased interest, several bioinformatic tools are available. However, each tool has its advantages and disadvantages, which makes the choice of the most appropriate tool for bench scientists and clinicians difficult. Here, we have benchmarked bioinformatic tools to detect RES from RNA-seq data. We provide a comprehensive view of each tool and its performance using previously published RNA-seq data to suggest recommendations on the most appropriate for utilization in future studies.

## 1. Introduction

RNA editing is a widespread post-transcriptional modification in eukaryotes that includes several types of biochemical changes such as deletions, insertions, and base conversions [1]. Adenosine-to-inosine (A-to-I) RNA editing is the most prevalent type of RNA editing in metazoans, and is catalyzed by adenosine deaminase acting on the RNA (ADAR) enzyme [2], which acts on double-stranded RNA (dsRNA). There are three ADAR genes in the mammalian genome: (1) ADAR1, which encodes for two protein isoforms, ADAR1p150 and ADAR1p110; (2) ADAR2; and (3) the catalytically inactive ADAR3 [2,3]. In addition to A-to-I editing, there is also cytidine-to-uridine (C-to-U) RNA editing, which is catalyzed by apolipoprotein B and mainly found in plants, although it is also present in mammals [4].

Among the RNA modifications that have been identified to date [5], A-to-I and C-to-U RNA editing are the only ones that can be readily detected from RNA sequencing (RNA-seq) data since the detection of other RNA modifications requires chemical modifications (e.g., bisulfite RNA-seq to detect 5-methylcytosine (m5C) sites [6,7]) or immunoprecipitation before the sequencing library preparation (e.g., N6-methyladenosine sequencing (m6A-seq) [8]). RNA editing sites (RES) can be identified by comparing matching RNA-seq and DNA sequencing (DNA-seq) [9]. In this approach, RES can be identified as adenosine to guanine (A > G) or cytidine to thymidine (C > T) mismatches for A-to-I or C-to-U RNA editing, respectively [10]. However, this approach may not be cost-effective or feasible due to the requirement of matching RNA- and DNA-seq data from the same sample of cells or tissues. To circumvent this problem, computational strategies to detect RES from RNA-seq data alone have been developed. To correctly identify RES using these approaches, several filters must be implemented within the associated pipeline to reduce the number of false positives. These filters are designed to minimize the presence of sequencing errors and known nucleotide differences among individuals (e.g., single nucleotide polymorphisms (SNPs)) [11]. Additionally, the usage of strand-specific RNA-seq protocols is preferred as it facilitates the identification of RES in regions with overlapping transcripts generated from opposite strands [12,13].

With over one million predicted A-to-I RNA editing sites in human cells [14], the biological effects of RNA editing are diverse, including apoptosis and cell survival [15,16], neural functions [17,18], and immune responses [19,20], such as the recognition of self to non-self dsRNA [21]. Given that RES can be readily detected from RNA-seq data, interest in analyzing RNA editing patterns has significantly increased in recent years [22]. To cope with such increased demands, there are several bioinformatic tools available with different approaches for the identification and prediction of RES. Here, we review bioinformatic tools to provide a comprehensive view of RES detection. To maximize the extraction of biologically relevant results from RNA-seq data, we benchmarked the performance of some of the available tools using published RNA-seq data of human cells with ADAR1 ablation. By comparing the known and predicted RES, we make specific recommendations for choosing the best-performing tool with the most straightforward installation, including associated software packages.

## 2. Materials and Methods

### 2.1. Computational Environment

This study was conducted on a multi-user Linux server with 64 central processing units (CPU) and 528 gigabytes (GB) of random-access memory (RAM). For each tool tested, the run time, percentage of CPU usage, and maximum resident set size (RSS, which refers to the largest amount of physical memory a process has used) were clocked and measured.

### 2.2. RNA-Seq Dataset

The RNA-seq data used for the assessment of the five benchmarked tools were obtained from the Gene Expression Omnibus (GEO) databases, accession number GSE99249 [23]. The RNA-seq data were generated in triplicates from (1) wildtype human embryonic kidney 293T (HEK 293T) cells (defined as WT) (Sequence Read Archive (SRA) accession numbers: SRR5564274, SRR5564275, and SRR5564276); and (2) ADAR1 knockout HEK 293T cells generated using CRISPR/Cas9 system (defined as ADAR1KO) (SRA accession numbers: SRR5564272, SRR5564273, and SRR5564268). This dataset was chosen based on the assumption that ADAR1 is the major RNA editing enzyme with the most targeted bases.

### 2.3. Reference Genome and Annotations

Due to the requirements of the five benchmarked tools (RED-ML [24], SPRINT [25], REDItools2 [26], JACUSA2 [27], and BCFtools [28]), two different human assembly releases were used: GRCh37 and GRCh38. The most recent release for each reference genome and general gene transfer format (GTF) (release 19 and 43, respectively) file were downloaded from the GENCODE project website (https://www.gencodegenes.org/human/ (accessed on 7 February 2023)) [29]. Additionally, each tool has specific requirements regarding annotation files. RED-ML requires release 138 of the dbSNP database (http://hgdownload.soe.ucsc.edu/goldenPath/hg19/database (accessed on 15 February 2023)), the browser extensible data (BED) file of simple repeats (http://hgdownload.soe.ucsc.edu/goldenPath/hg19/database (accessed on 15 February 2023)) from the human assembly release 19 (GRCh37), and the BED file with Alu sequences (http://hgdownload.soe.ucsc.edu/goldenPath/hg19/database (accessed on 15 February 2023)) from the same human assembly. SPRINT provides a Python script to adapt a Repeat Masker (rmsk) file into a BED file format. The rmsk file was downloaded from UCSC Table Browser (http://genome.ucsc.edu/cgi-bin/hgTables (accessed on 17 February 2023)) using the human assembly release GRCh38.

### 2.4. Reads Processing and Mapping

FASTQ files were downloaded using the SRA toolkit (v3.0.1) (https://github.com/ncbi/sra-tools/wiki/01.-Downloading-SRA-Toolkit (accessed on 3 February 2023)). Before mapping sequencing reads, the last 20 bases of FASTQ files were trimmed using fastx_trimmer (FASTX Toolkit 0.0.13) [30] to remove low-quality bases. Trimmed FASTQ files were aligned to two different human reference genome assemblies (GRCh37 and GRCh38) using three different RNA-seq aligners: the non-splice-aware Burrows–Wheeler Aligner (BWA) (v 0.7.17; mem algorithm) [31], and the splice-aware aligners HISAT2 (v 2.2.1) [32] and STAR (v 2.6.0a) [33]. For the BWA, one of the benchmarked tools, RED-ML, requires the use of a reference file which combines the reference genome of the human assembly release 19 (GRCh37) with exonic sequences surrounding known splice junctions [34]. To comply with this requirement, JAGuaR [35] was employed, which is an alignment protocol for paired-end RNA-seq reads that uses BWA to align reads to the genome and reference transcript models, including exon–exon junctions. Using SAMtools (v 1.16) [28] and Picard Tools (v 1.119) [36], the generated BAM files were processed to remove duplicates and include only mapped and properly paired reads with a minimum alignment quality score of 20 for further analysis.

### 2.5. Analysis with RNA Editing Detection Tools

To benchmark the performance of RNA editing detection tools, the following tools were tested: RED-ML [24], SPRINT [25], REDItools2 [26], JACUSA2 [27], and BCFtools [28] (v 1.16.1). Of note, although REDItools2 offers two modes of analysis (i.e., the serial and the parallel mode), we only tested the serial version in this study as the parallel mode version may not be common for all users. The processed BAM files were analyzed individually for each tool, except for JACUSA2, which requires sample replicates when only RNA-seq data are employed. The options used for each tool can be found on our GitHub page: https://github.com/davidrm-bio/Benchmark-of-RNA-Editing-Detection-Tools/tree/main/Tools (accessed on 12 July 2023). Some of the benchmarked tools require a pre-processing step. For example, SPRINT requires changing the MAPing Quality (MAPQ) values of Sequence Alignment MAP (SAM) files for non-BWA generated BAM files, which can be performed with a Python script provided by the authors. JACUSA2 requires the MD tags (defined as string-encoding mismatched and deleted reference bases, such as SNPs) of BAM files to be populated, which can be undertaken with SAMtools.

BAM files were processed separately in a similar way for all five tools tested. Results of BCFtools, RED-ML, SPRINT, and REDItools2 were processed individually for each condition (i.e., WT and ADAR1KO). It should be noted that although SPRINT provides a Python script to process non-BWA generated SAM files, the pre-processing of SAM files generated with STAR was not successful, which was noted by the authors [25]. Thus, the analysis with STAR for SPRINT was excluded. Due to the lack of DNA-seq data for the samples analyzed, we removed known variants using a list of SNPs in variant call format (VCF) file for the HEK 293T cell line, which is available at http://hek293genome.org/v2/about.php (accessed on 2 June 2023) [37]. The BAM files from BCFtools, SPRINT, and REDItools2 were generated by mapping to the human assembly GRCh38 and the BAM files from RED-ML were generated by mapping to the human assembly GRCh37. Thus, the VCF file was adapted to both the human assembly GRCh37 and GRCh38. The results of JACUSA2 were generated by mapping to the human assembly GRCh38 and were processed with the R package JACUSA2helper, keeping only variants present in the three replicates. The corresponding VCF file was also employed to remove known SNPs. To facilitate the comparison between the five different tools, the results of BCFtools, RED-ML, SPRINT, and REDItools2 were further processed by keeping only variants present in the three replicates. In addition to removing known SNPs for the HEK293T cell line, the following parameters from the output files were used in the downstream processing. The results from SPRINT, REDItools2, and JACUSA2 were filtered according to the number of reads that supported the variants, using different thresholds, including 2, 4, 6, 8, and 10. In the case of RED-ML, the number of supporting reads is not provided. Instead, the detection threshold is provided, which was interpreted as the probability of being a RES. Several thresholds were employed, including 0.5, 0.6, 0.7, 0.8, and 0.9. Lastly, for BCFtools, different threshold values for the minor allele frequency (MAF) were used (i.e., none and 0.1). After processing results for the different tools, the total number of A-to-I editing events as the sum of A > G and T > C were recovered. The total number of RES was compared to the registered data in REDIportal [38], a database of A-to-I editing events in humans. Additionally, we recovered the total number of known SNPs reported by each tool [38].

## 3. Results

### 3.1. Availability of RNA Editing Detection Tools

In total, 10 major RNA editing detection tools were compared according to the following criteria: required formats for input files, handling of strandedness and/or replicate samples, and the availability of read mappers or usage of the specific mappers (Appendix A). A detailed overview of the features of each tool is provided in the following text.

REDItools [39] was the earliest RNA editing detection tool to be introduced in the field and can be used for both the detection of known RES registered in RNA editing databases (e.g., REDIportal [38]) and the de novo prediction of RES without requiring a priori RNA editing information. This can be achieved either by using RNA-seq data alone or by comparing RNA- and DNA-seq data. If matching DNA-seq data are not available, variants are compared to an empirical substitution distribution and only statistically significant positions are reported. This tool takes pre-aligned reads in BAM format as the input. Several filters and quality checks can be applied to remove positions according to different parameters, such as base and mapping quality score, position coverage, and the removal of substitution in homopolymeric regions (i.e., regions that include stretches of the same nucleotide (e.g., AAAAA or TTTTT)). The same authors modified REDItools by introducing an optimized and parallel multi-node version, REDItools2 [26]. REDItools2 provides two modes of analysis: (1) a serial and (2) a parallel mode, which requires the installation of a message-passing interface (MPI) implementation.

GIREMI [40] identifies RES from RNA-seq data alone based on allelic linkage [41]; that is, the tendency of genetic markers physically near to each other to be inherited together during meiosis chromosomal crossover. Two nearby SNPs should maintain the same haplotype, while RES are likely to vary among reads relative to nearby SNPs. Based on this assumption, this method calculates the mutual information between publicly available SNP sites (such as those found in the dbSNP database [42]) and uncharacterized RNA variants in the RNA-seq sample. In addition to allelic linkage, a generalized linear model was employed to enhance the predictive power. GIREMI takes BAM files and a list of filtered single nucleotide variants (SNVs) as the input and if several BAM files are provided, they are treated as replicates and are combined into one dataset. The list of SNVs must include both SNVs available in databases (e.g., dbSNP database) and SNVs present in the RNA-seq samples, obligating users to perform genotype-calling analysis. Additionally, variant call format (VCF) files generated by variant callers must be adapted to the input format specified by the authors. Due to the increased demand for long reads to cover the full-length transcriptome (developed by Pacific Bioscience (PacBio) (in Menlo Park, California, United States of America) and Oxford Nanopore Technologies (ONT) (in Oxford Science Park, Oxford, UK)), the same laboratory developed a new method called L-GIREMI [43], which uses a similar approach as GIREMI for the prediction of RES from long-read RNA-seq data.

RES-Scanner [44] performs genome-wide identification and annotation of RES for any species, although its usage is limited by the availability of matching RNA- and DNA-seq data for the same sample, which is not always possible. The annotation of RES is only possible if annotation files with relevant features are provided. RES-Scanner uses a combination of different statistical models (e.g., Bayesian, binomial, and frequency) for homozygous genotype calling and filters (e.g., mapping quality, removal of duplicates, and depth of reads) to remove potential false-positive RES. RES-Scanner uses either raw reads in FASTQ format or pre-aligned reads in BAM format as the input. In the former case, BWA [31] is used as a read mapper to align FASTQ files with a combination of the reference genome and exonic sequences surrounding known splice junctions. A new version of RES-Scanner, RES-Scanner2, is now available, which can identify hyper-editing sites; that is, regions with high editing levels due to the tendency of ADAR1 to edit sites in clusters [45,46].

RNAEditor [11] is a fully automated pipeline that takes uncompressed FASTQ files as the input and performs all the necessary steps to predict RES. This tool has both a command line and a graphical user interface (GUI) version. The approach of RNAEditor consists of three steps: (1) read mapping via BWA; (2) SNP calling and purification via the GATK tool [47]; and (3) the annotation of RES. To reduce the number of false-positive RES, known SNP sites are filtered out using the information gathered from the dbSNP database, 1000 Genomes Project [48], and HAPMAP project [49]. Additionally, a clustering algorithm is used to detect highly edited regions, termed editing islands, because ADAR1 catalyzes deamination in clusters of dsRNA sequences. RNAEditor has been updated to support more recent versions of some dependent software programs, including Python3 and PyQt5, although it still requires an old version of GATK (v3.7) and Java (v8), obligating users to downgrade these dependent software products. Furthermore, RNAEditor uses discontinued datasets, such as the HAPMAP project (last release: Ensemble 97). Thus, compatibility with the existing and most up-to-date operating systems and programs is low.

JACUSA [50] detects SNVs by comparing RNA-DNA or RNA-RNA sequencing samples and integrating information from replicate experiments. The identification of position-specific RES using RNA- and DNA-seq data is straightforward, as RNA editing is a post-transcriptional modification that is not present in genomic DNA sequences. For RNA-RNA comparisons, RNA-seq data from two different conditions (e.g., control and knockdown samples) are compared to identify differential RES. In addition, JACUSA removes commonly known artefacts such as those produced by mapping programs or sequencing technologies. JACUSA2 [27], its successor, has a shorter running time and captures more complex read signatures, including substitutions, insertions, deletions, and read truncations. Additionally, it includes a new mode of analysis to detect read arrest events in pair-end read samples. Read arrest events lead to shorter reads and can happen during the library preparation due to the premature termination of the reverse transcriptase because of RNA degradation or structures or the presence of pseudouridines [51]. Both JACUSA and JACUSA2 use BAM files as the input. JACUSA2 is implemented with a complementary R package called JACUSA2helper. This R package was developed to assist with the downstream processing of JACUSA2 results, such as filtering and plotting. For example, JACUSA2helper removes the sites that have been marked by JACUSA2 as artefacts. It also filters sites according to their coverage and variants.

SPRINT [25] identifies RES and hyper-editing sites without any pre-processing step to remove known SNPs. This is advantageous because not all SNP sites should be considered as false-positive RES, as individual differences (e.g., one person compared to another) may be present on each SNP site. The analysis of SPRINT consists of three steps: (1) read processing; (2) SNV calling; and (3) the identification of RES. The last step is performed by clustering SNV duplets, which are defined as consecutive SNVs that have the same type of variation, such as two consecutive SNVs corresponding to A > G differences in RNA-seq reads compared to the reference genome. The assumption here is that a pair of consecutive SNVs tend to be true RES if they are within 400 nucleotides (nt) from each other, while a pair of consecutive SNVs tend to be SNPs if they are within 1600 nt. These assumptions are because ADAR enzymes tend to act in clusters. SPRINT can take uncompressed FASTQ files or BAM files as the input. In the former case, BWA is used as a read mapper. It should be noted that the identification of hyper-editing sites is only available if raw FASTQ files are used for the analysis.

RESIC [52] can classify and identify both RES and hyper-edited regions for any organisms and any number of input datasets by using an alignment graph model and multiple filtering steps. This tool accepts uncompressed FASTQ files as the input, although the usage of RESIC is not very convenient, as users cannot provide file names as command options, but must manually edit the Python script specifying the input data. Optionally, users can specify a negative dataset, i.e., FASTQ files that do not exhibit the desired editing phenomena, such as those produced in RNA-seq experiments from cells with inactivated ADAR genes (i.e., ADAR ablated samples). Negative datasets are used to exclude changes associated with SNPs. If negative datasets are not available, a list of SNPs in variant call format (VCF), such as those available in the dbSNP database, can be used. However, negative datasets and the VCF file from the dbSNP database cannot be provided at the same time. In the first part of the analysis, FASTQ files are aligned to the reference genome using bowtie [53] as a read mapper. Then, a graph aligner model is used to identify and classify RES into different categories (e.g., non-repetitive and hyper-non-repetitive A to C). During the identification of RES, several filters are employed, such as the removal of ambiguous reads and low-coverage sites.

Machine learning (ML) is increasingly being adopted within computer science and genomics and several ML-based tools for RES detection have been developed in recent years. For example, RDDpred [54] is the first pipeline developed based on an ML approach for the prediction of RES. RDDpred uses the information from RNA-editing databases (i.e., RADAR [55] and DARNED [56]) and the mapping errors set (MES) method [57] to generate a positive and a negative training set of RES, respectively. The MES method identifies the regions that are likely to generate SNP site errors by simulating randomly mutated sequencing reads that are aligned to the reference genome and analyzed using variant callers. In the first part of the analysis, positive and negative RES are identified using these training sets, which are then employed in the next part of the analysis using the prediction model. The prediction model, which is a random forest predictor, analyses the remaining sites to predict RES that are not registered in the RNA editing databases. RDDpred requires BAM files and several software programs (e.g., SAMtools and BCFtools [28], WEKA [58], and BAMtools [59]). If several BAM files are provided, they are treated as replicates. To maximize reproducibility, RDDpred includes these external software programs within the package, making its installation easier. However, most of these are outdated versions and may require the re-compilation of the source code in some systems. It should be noted that the information provided to predict RES (i.e., from the positive and negative datasets) was generated using the old human assembly release 19 (GRCh37) and has not been updated. In addition, BAM files must be generated using the same version of the human assembly.

RED-ML [24] incorporates information from different features (e.g., reads, sequencing and alignment artefacts) and the properties of RES (i.e., the sequence context, whether the candidate site is in Alu regions, and the editing type) in order to make predictions using a logistic regression classifier to identify RES in a genome-wide manner. If DNA-seq data are available, SNPs can be specified from DNA-seq data using variant callers (e.g., GATK) and incorporated into the analysis. Although RED-ML takes BAM files as the input, a special reference should be employed for the BAM files generated from the BWA read aligner. This reference file should combine the human assembly release 19 (GRCh37) and exonic sequences surrounding all known splice junctions. The usage of this tool is constrained by several limitations, such as its exclusive applicability to human RNA-seq data, dependency on BAM files generated using the outdated human assembly release 19 (GRCh37), and the ability to identify only those sites with relatively high editing levels.

DeepRed [60] uses deep learning and ensemble learning to directly identify candidate RES. To perform the analysis, this tool only needs a list of SNVs that contain information about the chromosomal positions, the reference allele, and the alternative allele. For this purpose, a processing step with variant callers (e.g., GATK of BCFtools) is mandatory, although there is no requirement for filtering steps or annotation. One benefit of this tool is its ability to process RNA-seq datasets from various species. Although DeepRed has relatively few dependent software requirements, it is dependent on MATLAB, which requires a paid license. Due to this reason, we did not consider this tool in our benchmarking analysis.

In addition to the above-mentioned stand-alone tools, there are also several online resources for detecting RES, such as DREAM (Detection of RES associated with miRNAs) [61], AIRlINER (Assessment of editing sites in non-repetitive regions) [62], and REP (Prediction of editing sites and their effect in humans) [63]. The advantage of using online tools is that the users do not need high computational power and disk space, although the uploading time and the capacity of handling multiple datasets can be challenging.

Lastly, there are databases for RES. The two previously mentioned databases, RADAR (last updated in 2014) and DARNED (last updated in 2012), include RES for humans, mice, and fruit flies. Other databases are (1) REDIdb (last updated in 2018) [64] for plant organellar genomes; (2) REDIportal (last updated in 2020) [38] for humans and mice; (3) PED (last updated in 2019) [65] for plants; (4) EDK (last updated in 2018) [66] for disease-related editing events in humans; and (5) TCEA (last updated in 2018) [67] for cancer-related editing events. One should note that many of the tools and databases mentioned above have been outdated for several years, which calls for major updates for these tools and the introduction of newer tools for detecting RES and categorizing the identified RES in a database format.

### 3.2. Comparison of Benchmarked RNA Editing Detection Tools

As listed in the previous section and summarized in Appendix A, there are numerous bioinformatic tools available to identify RES from RNA-seq data as well as by comparing DNA- and RNA-seq data from the same sample. Given that the RNA-seq technique is readily used as a screening tool and that research interest in epitranscriptomics has been growing rapidly in various fields of science, here we benchmarked several of these tools. The selection criteria were based on both the ease and compatibility of installation and configuration on the Linux server, including dependent software products, which in some cases require the user to downgrade to old and outdated versions, and the ease and convenience of usage. We included the following tools for further analysis: RED-ML, SPRINT, JACUSA2, and REDItools2. In addition to these tools, we utilized BCFtools for genotype calling, in order to evaluate the efficacy of this straightforward approach in comparison to the other tools.

As discussed in the previous section, each tool has different input requirements, although most of the benchmarked tools work directly on BAM files. While there is little restriction on the aligner used for generating these BAM files, some tools have specific recommendations and require pre-processing steps of BAM files, as they are optimized for specific aligners and parameters. For example, RED-ML is optimized for BWA and TopHat2 [68] and if BWA is used, a reference file must be created by combining the reference genome of the human assembly release 19 (GRCh37) and exonic sequences surrounding known splice junctions [34]. Both SPRINT and JACUSA2 require a pre-processing step for BAM files. For SPRINT, this requires changing the MAPQ values of SAM files for non-BWA-generated BAM files. However, SPRINT is not recommended to be used for splice-aware aligners as the authors report that it would require modifications to the current workflow [25]. JACUSA2 requires the MD tags of BAM files to be populated.

For the assessment of the four selected tools (RED-ML, SPRINT, JACUSA2, and REDItools2), the most recent version of each tool was installed by cloning the GitHub repository (the version used can be found in Table 1). To assess the performance of these four tools and BCFtools, the published RNA-seq data of ADAR1 ablated cells (denoted ADAR1KO) was compared to the control wildtype cells (denoted WT) (GEO, accession number GSE99249). Although the original study [23] includes further RNA-seq libraries (e.g., ADAR1p150 ablated cells as well as all cell lines treated with interferon beta to induce the activities of ADAR enzymes), we focused on these two conditions (WT and ADAR1KO) with the assumption that the ADAR1 enzyme is the major RNA editing enzyme, and thus very few editing events should be observed in the ADAR1KO samples. The interferon-induced data were not employed to avoid overestimating the naturally occurring RES. For each condition, three replicates were analyzed.

As it is well known that the mapping rate of each read mapper differs significantly [69], trimmed FASTQ files were aligned to the human reference genome using three different aligners—BWA, HISAT2, and STAR—to compare the effects of the choice of RNA-seq aligner in the prediction of RES. Due to the requirement of RED-ML to use the human assembly release 19 (GRCh37), FASTQ files were aligned to both the human assembly release GRCh37 (used for RED-ML) and GRCh38 (used for BCFtools, SPRINT, JACUSA2, and REDItools2). Appendix A shows the total number of reads present in raw FASTQ files as well as the average number of reads present in BAM files for each human assembly release after processing, removing duplicates, and keeping only mapped and properly paired reads with a minimum alignment quality score of 20.

Together with the run time required to analyze all six samples tested, the central processing unit (CPU) usage and physical memory (random-access memory (RAM)) usage are important considerations when analyzing RNA-seq data, especially if there are multiple samples per condition (Table 1). As expected, these values varied according to the tool used for the analysis. For example, REDItools2 showed the longest run time (~9 days) but one of the lowest usages of physical memory. On the other hand, JACUSA2 showed the fastest run time (3.7 h), due to the usage of multiple threads, but had the highest usage of physical memory. RED-ML and SPRINT showed similar usages of memory, although the analysis with RED-ML required significantly more time (63.77 h compared to 23.33 h, respectively). The high run time displayed by REDItools2 and RED-ML can be partly explained by the approach employed by these tools, as they both include an mpileup step, which is noted to be the bottleneck in the analysis by the authors of RED-ML [24]. Another factor to consider is that these analyses were run on a multi-user server, and therefore the load of the system might have led to the overestimation of the run time. Due to significant variations in the required time (from hours to days), users are recommended to base the choice of RNA editing detection tool on to the size of the dataset in their study.

Since BCFtools, RED-ML, SPRINT, and REDItools2 do not accept replicate samples, the replicates for each condition (i.e., WT and ADAR1KO) were analyzed and processed individually, including removing variants present the HEK 293T cell line. Each output file generated by the different benchmark tools provides different parameters that can be used in the downstream processing to identify true RES. For example, RED-ML provides a detection threshold that provides information about the confidence in the prediction. REDItools2 and SPRINT provide the number of reads that support RES, while the minor allele frequency (MAF) can be used to remove variants with low frequency for BCFtools. We chose different values for each of the following parameters to assess how they impact the prediction of RES: (1) 2, 4, 6, 8, and 10 for the number of supporting reads; (2) 0.5, 0.6, 0.7, 0.8, and 0.9 for the detection threshold; and (3) none and 0.1 for the MAF. In general, increasing the threshold values of these parameters led to a decrease in the number of RES detected for BCFtools (Appendix A), RED-ML (Appendix A), and REDItools2 (Appendix A). This decrease in the prediction of RES is accompanied by an increase in the fraction of RES supported by the REDIportal database. In contrast, the fraction of RES in the REDIportal database barely changes for SPRINT (Appendix A). In the case of SPRINT, almost all the RES detected using the lowest threshold value (i.e., 2 supporting reads) were supported by REDIportal, and increasing this threshold only led to the exclusion of RES. Another factor of interest is the correlation observed between the fraction of RES in Alu regions and in the REDIportal database, which might be explained because of the fact that 90.66% of the RES registered in REDIportal are located within Alu regions. As expected, in all the cases the number of predicted RES as well as the fraction of them present in the REDIportal database for ADAR1KO samples were significantly lower compared to WT samples, supportive of a considerable fraction of RES being mediated by ADAR1.

Since, as discussed above, higher parameter thresholds can improve the quality of RES predictions, a post-filtering step is recommended to exclude possible false positives. However, since the effects of adjusting these parameters vary according to the specific tools, as a means to compare them the lowest threshold value for each tool was used as a filter (i.e., a threshold value of 2 for the number of supporting reads in REDItool2 and SPRINT; a threshold value of 0.5 for the detection threshold in RED-ML; and all RES reported by BCFtools).

SPRINT showed the highest prediction of RES supported by the REDIportal database, although the lowest number of predicted RES (Figure 1). The reason for the low number of reported RES, especially in ADAR1KO samples, could be explained by the fact that the SPRINT workflow relies on the assumption that SNV duplets (i.e., a pair of consecutive SNVs) tend to be true RES if they are within 400 bp of each other, because ADAR enzymes tend to act in clusters. Additionally, these tool clusters predicted RES and only report clusters of a specific size [25], constraining predictions to regions with high levels of RNA editing. In general, the majority of the RES predicted by each tool are located in Alu regions. These simple repeats are complementary and can form dsRNA, which is the preferred substrate for ADAR enzymes [70]. Contrary to the detection of RES in non-Alu regions, the detection in Alu regions is straightforward, as it has been observed that almost all adenosines in Alu regions that form dsRNA structures are edited [14].

In accordance with the previous report [71], BWA and STAR tended to predict the highest number of RES, while the lowest number of RES was identified with HISAT2. Splice-aware RNA-seq aligners consider the spliced nature of RNA and are, therefore, believed to work better in detecting sequence variants [72]. However, our results suggest that the non-splice-aware aligner BWA predicted similar and in some cases even more RES than the splice-aware RNA-seq aligners, although better support in the REDIportal database was observed with splice-aware aligners. Of note, the differences observed in HISAT2 and STAR can be explained by alignment algorithm differences [73].

Contrary to the four tools discussed above, the analysis with JACUSA2 was performed by combining the three replicates into one sample, due to the requirement of using replicate samples if RNA-seq data alone are analyzed with JACUSA2. RES present in the HEK 293T database were excluded from the analysis. Thus, to compare the five benchmarked tools, we re-analyzed the results of the tools by keeping only RES present in the three replicates. JACUSA2, like the other tools discussed before, provided several parameters in the output files which could be used for downstream processing, such as tags indicating if variants were artefacts. Like REDItools2 and SPRINT, JACUSA2 output files also include the number of supporting reads for the variants. The same threshold values employed with REDItools2 and SPRINT for this parameter (i.e., 2, 4, 6, 8, and 10) were used with JACUSA2. In contrast to what was observed before, increasing the number of supporting reads did not result in a decrease in the total number of RES and the fraction of RES present in REDIportal (Appendix A). As in Figure 1, we chose the same threshold values for the different parameters mentioned before filtering the re-analyzed results (i.e., a threshold value of 2 for the number of supporting reads; a threshold value of 0.5 for the detection threshold; and all RES reported by BCFtools). Combining replicates into one sample significantly reduced the number of predicted RES (Figure 2); however, this decrease was accompanied by an increase in the confidence of the predicted RES. In this approach, JACUSA2 and REDItools2 displayed the best results, with similar counts for the splice-aware aligners, although REDItools2 had slightly better support in the REDIportal database. For the other tools, similar tendencies as before were observed, with RED-ML and SPRINT having the lowest counts but with good support in the REDIportal database.

To effectively identify RES, it is recommended to use DNA-seq and RNA-seq data from the same samples, as this allows the exclusion of SNPs. However, studies re-analyzing publicly available RNA-seq data may encounter issues with this approach, as most published datasets do not include DNA-seq data, as they can be costly to generate. In this case, publicly available SNPs (e.g., those in the dbSNP database) can be used to overcome this issue. However, this approach has some limitations as it can lead to the exclusion of SNPs that are not present in the sample. In this study, no DNA-seq data was available in the original study generating the RNA-seq data. To overcome this issue and incorporate genomic information, we removed SNPs present in the HEK293T cell type, based on an available database. As shown in Figure 3, BCFtools, followed by REDItools2, predicted the highest amount of RES which correspond to true SNPs. This indicates that a post-processing step to remove known SNPs is advisable for these tools. To this end, SPRINT is the most effective in avoiding the inclusion of SNPs, followed by JACUSA2 and RED-ML. If DNA-seq data are not available, these tools would therefore be more suited to minimize the inclusion of false positive RES.

## 4. Discussion

Although a similar comparative study was published previously [71], no update has been made since its publication in 2019, which excluded tools published more recently (e.g., RED-ML and SPRINT). Additionally, some of the tools have been updated (e.g., REDItools2 and JACUSA2). To this end, we comprehensively reviewed the currently available RNA editing detection tools. As RNA editing events can be readily detected from RNA-seq data, we especially emphasized the ease of installation and run time to help select the best tool for the users’ needs. In addition, we benchmarked four tools (RED-ML, SPRINT, JACUSA2, and REDItools2) in comparison to the simple variant call tool, BCFtools, to demonstrate the performance of each tool by using the results obtained from three popular read aligners—BWA, STAR, and HISAT2.

The five benchmarked tools were assessed using two different approaches: (1) individual analysis and (2) combining replicate samples into one. If replicate samples are available, an additional layer of information can be added during the detection of RES as artefacts generated during library preparation can be excluded. However, most of the tools do not accept replicate samples and individual analysis is mandatory. Only JACUSA2 accepts replicates, and therefore a downstream processing step to keep RES present in all or some of the replicates would be recommended to improve confidence. After the identification of RES, we recommend employing additional filters in the downstream processing of the output files, such as the number of supporting reads, to minimize the amount of false positive RES.

Based on the re-analysis of the published RNA-seq data of genetic deletion of the ADAR1 gene in human cells, we recommend STAR as the read aligner and REDItools2 as the RNA editing detection tool for the identification of RES, if the analysis time is not a constraint, as this combination identifies the second-largest number of RES in WT samples in both high and low stringencies of threshold cut-off levels in each analysis pipeline (Table 2 and Table 3). Although there tends to be more RES with BWA as a read aligner, STAR is much faster in aligning the reads than BWA and displays higher support in the REDIportal database. If users are interested in the more confident identification of RES, SPRINT, with its recommended aligner BWA, should be used as it displayed the highest support in REDIportal compared to the other tools, and outperforms the other tools in minimizing the inclusion of SNPs. Note, however, that SPRINT is more suited for the identification of RES occurring in clusters, which leads to a lower number of reported RES. If the analysis time is of the essence and replicate samples are available, we recommend the combination of STAR and JACUSA2.

## 5. Conclusions

RNA editing events are known to increase in cellular stress and in disease conditions [22,74,75], emphasizing the importance of their accurate detection. We hope that our comprehensive comparison of RNA editing detection tools in this study will assist readers in selecting the most appropriate tool for detecting RES from their RNA-seq data. Given the growing quantity of RNA-seq data available, our study has wide potential to positively influence analyses of RNA editing across a variety of contexts, such as healthy and diseased tissues. Furthermore, since it is well known that RNA editing affects the structures of RNA molecules [76], a promising future direction in this field could be to elucidate the impacts of RNA editing on changes in functional roles of long non-coding RNAs (lncRNAs). The above tools and strategies discussed in our study would be relevant in this context.

## Figures and Tables

**Figure 1 biotech-12-00056-f001:**
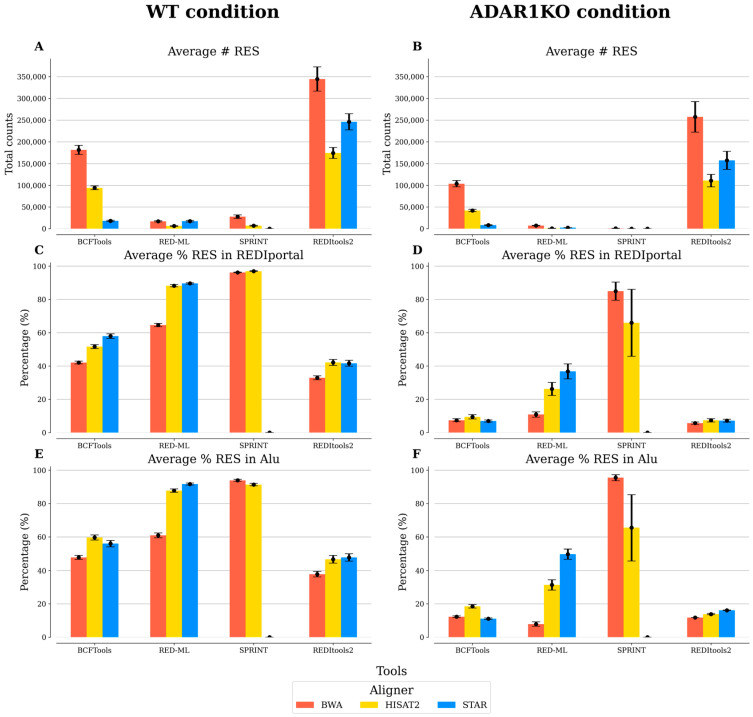
Comparison of the five benchmarked tools analyzed individually. For each condition, replicate samples (*n* = 3) were processed individually, keeping RNA editing sites (RES) absent in HEK293T cells. The average and standard error of the mean are reported for each one of the measurements included: (**A**,**B**) average number (#) of total RES; (**C**,**D**) average percentage (%) of RES in REDIportal; and (**E**,**F**) average percentage of RES in Alu regions. The results of BCFtools were filtered using no threshold value for the minor allele frequency. The results of RED-ML were filtered using a detection threshold value of 0.5. The results of SPRINT and REDItools2 were filtered using a threshold value of 2 for the number of supporting reads.

**Figure 2 biotech-12-00056-f002:**
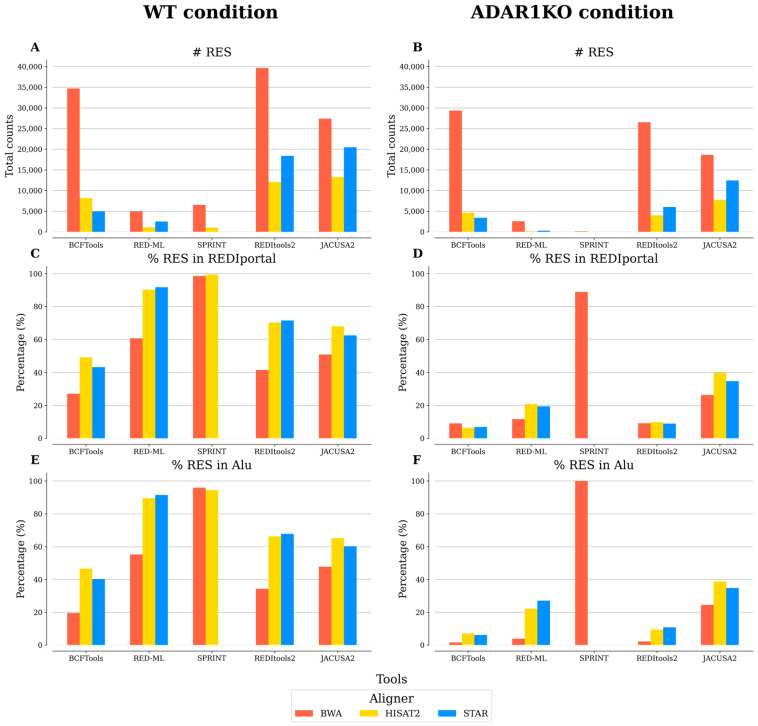
Comparison of the four benchmarked tools analyzed by merging replicates. For each condition, replicate samples (*n* = 3) were merged into one sample, excluding RNA editing sites (RES) not present in the three replicates and present in HEK293T cells. Different measurements are reported: (**A**,**B**) total number (#) of RES; (**C**,**D**) percentage (%) of RES in REDIportal; and (**E**,**F**) percentage of RES in Alu regions. The results of BCFtools were filtered using no threshold value for the minor allele frequency. The results of RED-ML were filtered using a detection threshold value of 0.5. The results of SPRINT and REDItools2 were filtered using a threshold value of 2 for the number of supporting reads.

**Figure 3 biotech-12-00056-f003:**
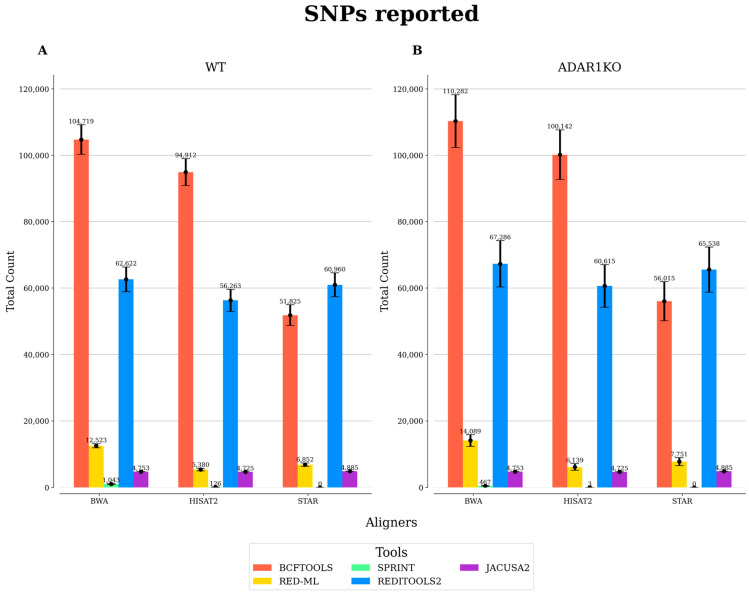
Number of SNPs reported by each RNA editing detection tool. The total number of SNPs was obtained using a list of known SNPs in the HEK293T cell line. Results from each tool were processed to count the total number of SNPs per sample. The average number of SNPs is reported for WT (**A**) and ADAR1KO (**B**) conditions in BCFtools, RED-ML, SPRINT, and REDItools2. For JACUSA2, the total number of reported SNPs is represented since the analysis with JACUSA2 was performed by combining the three replicate samples into one.

**Table 1 biotech-12-00056-t001:** The time required to analyze each data. The run time refers to the average between the different RNA-seq aligners for the 6 samples analyzed. The maximum resident set size (RSS) indicates the largest amount of physical memory (random-access memory (RAM)) the tool has used for the entire analysis.

Tool	Real Run Time (h)	CPU %	Maximum RSS (GB)
BCFtools	9.83	105	1.06
RED-ML	63.77	99	12.42
SPRINT	23.33	98	12.17
JACUSA2	3.70	559	32.57
REDItools2	215.18	99	1.29

**Table 2 biotech-12-00056-t002:** Numbers of RES with low stringencies for threshold values. The percentage of the RES in ADAR1KO in comparison to WT indicates a reduction in the number (#) of RES upon genetically ablating ADAR1, as expected. N/A represents not applicable.

Sample Condition	Aligner	Tool	# RES	% (ADAR1KO/WT)
WT	BWA	RED-ML	17,110	
ADAR1KO	BWA	RED-ML	7228	42.24
WT	HISAT2	RED-ML	6158	
ADAR1KO	HISAT2	RED-ML	918	14.91
WT	STAR	RED-ML	17,309	
ADAR1KO	STAR	RED-ML	2267	13.10
WT	BWA	REDItools2	344,646	
ADAR1KO	BWA	REDItools2	257,445	74.70
WT	HISAT2	REDItools2	174,481	
ADAR1KO	HISAT2	REDItools2	110,643	63.41
WT	STAR	REDItools2	246,040	
ADAR1KO	STAR	REDItools2	157,254	63.91
WT	BWA	SPRINT	27,707	
ADAR1KO	BWA	SPRINT	919	3.32
WT	HISAT2	SPRINT	6903	
ADAR1KO	HISAT2	SPRINT	58	0.84
WT	STAR	SPRINT	N/A	
ADAR1KO	STAR	SPRINT	N/A	N/A
WT	BWA	JACUSA2	27,388	
ADAR1KO	BWA	JACUSA2	18,606	67.93
WT	HISAT2	JACUSA2	13,252	
ADAR1KO	HISAT2	JACUSA2	7739	58.40
WT	STAR	JACUSA2	20,455	
ADAR1KO	STAR	JACUSA2	12,431	60.77

**Table 3 biotech-12-00056-t003:** Numbers of RES with low stringencies for threshold values. The percentage of the RES in ADAR1KO in comparison to WT indicates the reduction in the number (#) of RES upon genetically ablating ADAR1, as expected. N/A represents not applicable.

Sample Condition	Aligner	Tool	# RES	% (ADAR1KO/WT)
WT	BWA	RED-ML	3949	
ADAR1KO	BWA	RED-ML	962	24
WT	HISAT2	RED-ML	1976	
ADAR1KO	HISAT2	RED-ML	206	10
WT	STAR	RED-ML	4262	
ADAR1KO	STAR	RED-ML	396	9
WT	BWA	REDItools2	42,891	
ADAR1KO	BWA	REDItools2	33,648	78
WT	HISAT2	REDItools2	15,083	
ADAR1KO	HISAT2	REDItools2	8234	55
WT	STAR	REDItools2	21,734	
ADAR1KO	STAR	REDItools2	11,651	54
WT	BWA	SPRINT	1358	
ADAR1KO	BWA	SPRINT	50	4
WT	HISAT2	SPRINT	415	
ADAR1KO	HISAT2	SPRINT	3	1
WT	STAR	SPRINT	N/A	
ADAR1KO	STAR	SPRINT	N/A	N/A
WT	BWA	JACUSA2	20,048	
ADAR1KO	BWA	JACUSA2	14,941	75
WT	HISAT2	JACUSA2	9642	
ADAR1KO	HISAT2	JACUSA2	6342	66
WT	STAR	JACUSA2	14,681	
ADAR1KO	STAR	JACUSA2	10,029	68

## Data Availability

The data analyzed in this article are available in the Gene Expression Omnibus (GEO) database (accession ID, GSE99249). The code used is available on GitHub (https://github.com/davidrm-bio/Benchmark-of-RNA-Editing-Detection-Tools (accessed on 12 July 2023)).

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
