# Peer review of "Benchmarking RNA Editing Detection Tools"

_biotech, 2023, doi:10.3390/biotech12030056_

Round 1

Reviewer 1 Report

In this manuscript, the author compared different RNA editing tools and provide valuable recommendations. The article is well-written and minor revision is suggested. 

1. In section 3.1, the authors listed 10 major RNA editing detection tools. However, as they mention, some of these tools are outdated. To enhance the usefulness of this information for readers, it would be beneficial if the authors could include the published year of each tool as a reference.

2. The author mentions a similar comparative study that was published previously. It would be informative to discuss whether the same RNA editing tools behave differently over time. Have there been significant updates or improvements in these tools since the previous study was conducted?

3. In order to provide a comprehensive overview, the author could discuss potential future work in this area. For example, what aspects of RNA editing tools can be further improved upon? Offering insights into future developments in this field would be valuable for readers.

Author Response

In this manuscript, the author compared different RNA editing tools and provide valuable recommendations. The article is well-written and minor revision is suggested.

Response: Thank you very much for your praise and valuable comments, which are addressed as follows:

  1. In section 3.1, the authors listed 10 major RNA editing detection tools. However, as they mention, some of these tools are outdated. To enhance the usefulness of this information for readers, it would be beneficial if the authors could include the published year of each tool as a reference.

Response: The date of latest version of each tool is provided in Supplementary Table S1. The reason for providing such information as Supplementary Data is that Supplementary Table S1 contains several other information that it will not fit into the limited space provided in the main text.

  1. The author mentions a similar comparative study that was published previously. It would be informative to discuss whether the same RNA editing tools behave differently over time. Have there been significant updates or improvements in these tools since the previous study was conducted?

Response: The above sentence was modified as follow:

“Although a similar comparative study was published previously [71], no update has been made since its publication in 2019, which excluded tools published more recently (e.g., RED-ML and SPRINT). Additionally, some of the tools have been updated (e.g., REDItools2 and JACUSA2).”

  1. In order to provide a comprehensive overview, the author could discuss potential future work in this area. For example, what aspects of RNA editing tools can be further improved upon? Offering insights into future developments in this field would be valuable for readers.

Response: The following sentences were added to the Conclusions section:

“As RNA editing events are known to increase in the stressed cells and under diseases [22,74,75], we hope that the comprehensive comparison of RNA editing detection tools in this study allows readers to choose the right tool for detecting RES from RNA-seq data comparing diseased tissues compared to the healthy ones to shed the lights on the effects of RNA editing on pathogenesis of various diseases. Also, it is well known that RNA editing affects the structures of RNA molecules [76], it is of great interest to elucidate the impacts of RNA editing on changes in functional roles of long non-coding RNAs (lncRNAs).”

Reviewer 2 Report

Dear Authors,

Reviewer comments biotech-2529878

The manuscript entitled „Benchmarking RNA editing detection tools“ represents a useful study aimed at a comparison of online tools useful for detection of RNA-editing sites, namely adenosine to inosine (A-to-I) conversion catalyzed by adenosine deaminase acting on RNA (ADAR) enzymes. I am a plant biologist and an expert on plant proteomics, not on RNA editing and epitranscriptomics, espcially in humans; however, i have a few comments on the present manuscript below:

1/ Databases for RNA editing (lines 328-329): The authors wrote about the databases of RADAR and DARNED with last updates in 2014 and 2012, respctively. I think that some newer and up-datwed version should be available and that the newest versions should be cited in the review text!!

2/ In addition to Table 1 providing basic characteristics of online tools for the detection of RNA editing, I think taht an analogous overview on the three popular read aligners BWA, STAR, and HISAT2 should be added. STAR aligner which is finally recommended in combination with JACUSA2; however, no characteristics are provided in the manuscript.

3/ The part 4. Discussion was most probably omitted in the manuscript since only the heading 4. Discussion was left in the manuscript. Maybe the following part 4. Conclusions is quite long and should be divided into Discussion and Conclusions.

Formal comments on the text related to English language and style:

Abstract, line 24: Modify the words „for use“ to „utilization“ in the statement: „We provide a comprehensive view on each tool and its performance using previously published RNA-seq data to suggest recommendations on the most appropriate utilization in future studies.“

Introduction, line 51: Add the word „problem“ following the words „To circumwent this problém,…“

Line 285: Add „a“ preceding the words „…to generate a positive and a negative training set of RES, respectively.“

Line 429: Add a dot at the end of the statement ending with „…within Alu regions.“

Line 482: Add a comma following the words „if replicate samples are done in Figure 1,…“

Conclusions, line 534: Add a comma following the statement „If replicate samples are available,…“

Line 537: Add a comma both before and after the word „therefore“ in the statment „…and, therefore, a downstream processingstep…“

Final recommendation:  Accept after a minor revision.

Dear Authors,

Reviewer comments biotech-2529878

The manuscript entitled „Benchmarking RNA editing detection tools“ represents a useful study aimed at a comparison of online tools useful for detection of RNA-editing sites, namely adenosine to inosine (A-to-I) conversion catalyzed by adenosine deaminase acting on RNA (ADAR) enzymes. I am a plant biologist and an expert on plant proteomics, not on RNA editing and epitranscriptomics, espcially in humans; however, i have a few comments on the present manuscript below:

1/ Databases for RNA editing (lines 328-329): The authors wrote about the databases of RADAR and DARNED with last updates in 2014 and 2012, respctively. I think that some newer and up-datwed version should be available and that the newest versions should be cited in the review text!!

2/ In addition to Table 1 providing basic characteristics of online tools for the detection of RNA editing, I think taht an analogous overview on the three popular read aligners BWA, STAR, and HISAT2 should be added. STAR aligner which is finally recommended in combination with JACUSA2; however, no characteristics are provided in the manuscript.

3/ The part 4. Discussion was most probably omitted in the manuscript since only the heading 4. Discussion was left in the manuscript. Maybe the following part 4. Conclusions is quite long and should be divided into Discussion and Conclusions.

Formal comments on the text related to English language and style:

Abstract, line 24: Modify the words „for use“ to „utilization“ in the statement: „We provide a comprehensive view on each tool and its performance using previously published RNA-seq data to suggest recommendations on the most appropriate utilization in future studies.“

Introduction, line 51: Add the word „problem“ following the words „To circumwent this problém,…“

Line 285: Add „a“ preceding the words „…to generate a positive and a negative training set of RES, respectively.“

Line 429: Add a dot at the end of the statement ending with „…within Alu regions.“

Line 482: Add a comma following the words „if replicate samples are done in Figure 1,…“

Conclusions, line 534: Add a comma following the statement „If replicate samples are available,…“

Line 537: Add a comma both before and after the word „therefore“ in the statment „…and, therefore, a downstream processingstep…“

Final recommendation:  Accept after a minor revision.

Author Response

The manuscript entitled „Benchmarking RNA editing detection tools“ represents a useful study aimed at a comparison of online tools useful for detection of RNA-editing sites, namely adenosine to inosine (A-to-I) conversion catalyzed by adenosine deaminase acting on RNA (ADAR) enzymes. I am a plant biologist and an expert on plant proteomics, not on RNA editing and epitranscriptomics, espcially in humans; however, i have a few comments on the present manuscript below:

1/ Databases for RNA editing (lines 328-329): The authors wrote about the databases of RADAR and DARNED with last updates in 2014 and 2012, respctively. I think that some newer and up-datwed version should be available and that the newest versions should be cited in the review text!!

Response: According to the information provided by the Database Commons, the year of last update for both tools are corrected as stated: https://ngdc.cncb.ac.cn/databasecommons/database/id/172(RADAR) and https://ngdc.cncb.ac.cn/databasecommons/database/id/338 (DARNED).

2/ In addition to Table 1 providing basic characteristics of online tools for the detection of RNA editing, I think taht an analogous overview on the three popular read aligners BWA, STAR, and HISAT2 should be added. STAR aligner which is finally recommended in combination with JACUSA2; however, no characteristics are provided in the manuscript.

Response: The basic characteristics of each tool are provided in Supplementary Table S1. The reason for providing such information as Supplementary Data is tha Supplementary Table S1 contains several other information that it will not fit into the limited space provided in the main text.

3/ The part 4. Discussion was most probably omitted in the manuscript since only the heading 4. Discussion was left in the manuscript. Maybe the following part 4. Conclusions is quite long and should be divided into Discussion and Conclusions.

Response: As suggested, the previous Conclusions section was now labeled as Discussion section. Also, the following Conclusions section as added:

“5. Conclusions

RNA editing events are known to increase in cellular stress and in disease conditions [22,74,75], emphasizing the importance of their accurate detection. We hope that our comprehensive compar-ison of RNA editing detection tools in this study will assist readers in selecting the most appro-priate tool for detecting RES from their RNA-seq data. Given the growing quantity of RNA-seq data available, our study has wide potential to positively influence analyses of RNA editing across a variety of contexts, such as healthy and disease tissues. Furthermore, since it is well known that RNA editing affects the structures of RNA molecules [76], a promising future direc-tion in this field could be to elucidate the impacts of RNA editing on changes in functional roles of long non-coding RNAs (lncRNAs). The above tools and strategies discussed in our study above would be relevant in this context.

Formal comments on the text related to English language and style:

Abstract, line 24: Modify the words „for use“ to „utilization“ in the statement: „We provide a comprehensive view on each tool and its performance using previously published RNA-seq data to suggest recommendations on the most appropriate utilization in future studies.“

Introduction, line 51: Add the word „problem“ following the words „To circumwent this problém,…“

Line 285: Add „a“ preceding the words „…to generate a positive and a negative training set of RES, respectively.“

Line 429: Add a dot at the end of the statement ending with „…within Alu regions.“

Line 482: Add a comma following the words „if replicate samples are done in Figure 1,…“

Conclusions, line 534: Add a comma following the statement „If replicate samples are available,…“

Line 537: Add a comma both before and after the word „therefore“ in the statment „…and, therefore, a downstream processingstep…“

Response: Thank you very much for critically reading our manuscript. All of the above parts have been corrected as suggested.

Reviewer 3 Report

Manuscript "Benchmarking RNA editing detection tools" is very interesting and important.

General comments:
Authors benchmarked bioinformatic tools to detect RES from RNA-seq data. Authors provided a comprehensive view of each tool and its performance using previously published RNA-seq data to suggest recommendations on the most appropriate for use in future studies.

Authors made specific recommendations for choosing the best-performing tool with the most straightforward installation, including associated software packages.

Detailed comments:
Line 105 - FASTQ - full name.
Figure 1: Complete with LSD or HSD values and homogeneous groups.
Figure 2: Complete with LSD or HSD values and homogeneous groups.

Paper needs minor revision.

Author Response

Manuscript "Benchmarking RNA editing detection tools" is very interesting and important.

General comments: Authors benchmarked bioinformatic tools to detect RES from RNA-seq data. Authors provided a comprehensive view of each tool and its performance using previously published RNA-seq data to suggest recommendations on the most appropriate for use in future studies.

Authors made specific recommendations for choosing the best-performing tool with the most straightforward installation, including associated software packages.

Response: Thank you very much for your praise and valuable comments, which we addressed below.

Detailed comments:

Line 105 - FASTQ - full name.

Response: The term, FASTQ, is not abbreviation.

Figure 1: Complete with LSD or HSD values and homogeneous groups.

Figure 2: Complete with LSD or HSD values and homogeneous groups.

Response: Could you kindly expand on "LSD" and "HSD"?

Reviewer 4 Report

Here are the comments : 

- Clearly explains the significance of studying RNA editing patterns and how it can be readily detected from RNA-seq data compared to other epitranscriptomic marks

- Provides a very comprehensive table summarizing numerous RNA editing detection tools currently available along with details about their features

- Benchmarking analysis of tools using real RNA-seq datasets makes it more practically useful for researchers

- Makes specific recommendations on choosing the optimal tool based on performance, ease of installation, computational requirements etc. 

- The introduction effectively sets the context and rationale for epitranscriptomic analysis of RNA editing

- Conclusions are clear, concise and actionable for researchers to select the best tool per their needs

- Manuscript is well-written and easy to understand even for non-computational biologists

- Comparing tools against RNA-seq data from ADAR1 knockout cells is a clever approach to benchmark performance

- Analysis of using different aligners (STAR, HISAT2, BWA) adds a useful dimension to compare tool performance

- Discussion of parameter thresholds and their effects on prediction results provides good insights

- Recommendations are supported by data and helpful for readers to select the optimal tool

- Figures are appropriate and convey the key results from the benchmarking analysis

- Overall, the manuscript makes a strong contribution that fits well with the journal's scope and epitranscriptomics section

Author Response

Here are the comments :

- Clearly explains the significance of studying RNA editing patterns and how it can be readily detected from RNA-seq data compared to other epitranscriptomic marks

- Provides a very comprehensive table summarizing numerous RNA editing detection tools currently available along with details about their features

- Benchmarking analysis of tools using real RNA-seq datasets makes it more practically useful for researchers

- Makes specific recommendations on choosing the optimal tool based on performance, ease of installation, computational requirements etc.

- The introduction effectively sets the context and rationale for epitranscriptomic analysis of RNA editing

- Conclusions are clear, concise and actionable for researchers to select the best tool per their needs

- Manuscript is well-written and easy to understand even for non-computational biologists

- Comparing tools against RNA-seq data from ADAR1 knockout cells is a clever approach to benchmark performance

- Analysis of using different aligners (STAR, HISAT2, BWA) adds a useful dimension to compare tool performance

- Discussion of parameter thresholds and their effects on prediction results provides good insights

- Recommendations are supported by data and helpful for readers to select the optimal tool

- Figures are appropriate and convey the key results from the benchmarking analysis

- Overall, the manuscript makes a strong contribution that fits well with the journal's scope and epitranscriptomics sectiona

Response: Thank you very much for your praise. To further improve our manuscript, the previous Conclusions section was now labeled as Discussion section. Also, the following Conclusions section as added:

“5. Conclusions

RNA editing events are known to increase in cellular stress and in disease conditions [22,74,75], emphasizing the importance of their accurate detection. We hope that our comprehensive compar-ison of RNA editing detection tools in this study will assist readers in selecting the most appro-priate tool for detecting RES from their RNA-seq data. Given the growing quantity of RNA-seq data available, our study has wide potential to positively influence analyses of RNA editing across a variety of contexts, such as healthy and disease tissues. Furthermore, since it is well known that RNA editing affects the structures of RNA molecules [76], a promising future direc-tion in this field could be to elucidate the impacts of RNA editing on changes in functional roles of long non-coding RNAs (lncRNAs). The above tools and strategies discussed in our study above would be relevant in this context.

Reviewer 5 Report

The article entitled " Benchmarking RNA editing detection tools". I see this manuscript as a research article, not a review. It provides a comprehensive view of different benchmarked bioinformatic tools to detect RES from RNA-seq data and its performance using previously published RNA-seq data to suggest the most appropriate for future studies. The topic is highly relevant and of general interest to the journal’s readers. I suggest accepting this manuscript for publication after minor editing.

·         The authors are highly recommended to avoid using a personal pronoun (e.g., We, our, etc.); they can use the third party in the past tense's passive voice.

·         The manuscript should be formatted as a research article, not a review article, so it should have a discussion section.

·         This manuscript has many website links for helpful information and/or tools, I suggest the authors make sure that all the links are active and working correctly. Some of them are not working, for example, line 96 and line 129.

Minor editing of English language required.